# Differential Modulation by Eicosapentaenoic Acid (EPA) and Docosahexaenoic Acid (DHA) of Mesenteric Fat and Macrophages and T Cells in Adipose Tissue of Obese *fa*/*fa* Zucker Rats

**DOI:** 10.3390/nu16091311

**Published:** 2024-04-27

**Authors:** Lena Hong, Peter Zahradka, Carla G. Taylor

**Affiliations:** 1Department of Food and Human Nutritional Sciences, University of Manitoba, Winnipeg, MB R3T 2N2, Canada; 2Canadian Centre for Agri-Food Research in Health and Medicine, St. Boniface Albrechtsen Research Centre, Winnipeg, MB R2H 2A6, Canada; pzahradka@sbrc.ca; 3Department of Physiology and Pathophysiology, University of Manitoba, Winnipeg, MB R3E 0J9, Canada

**Keywords:** obesity, adipose tissue, omega-3 fatty acids, α-linolenic acid, eicosapentaenoic acid, docosahexaenoic acid, T cells, macrophages, tumour necrosis factor-α, *fa*/*fa* Zucker rats

## Abstract

Polyunsaturated fatty acids (PUFAs) can alter adipose tissue function; however, the relative effects of plant and marine n3-PUFAs are less clear. Our objective was to directly compare the n3-PUFAs, plant-based α-linolenic acid (ALA) in flaxseed oil, and marine-based eicosapentaenoic acid (EPA) or docosahexaenoic acid (DHA) in high-purity oils versus n6-PUFA containing linoleic acid (LA) for their effects on the adipose tissue and oral glucose tolerance of obese rats. Male *fa*/*fa* Zucker rats were assigned to faALA, faEPA, faDHA, and faLA groups and compared to baseline *fa*/*fa* rats (faBASE) and lean Zucker rats (lnLA). After 8 weeks, faEPA and faDHA had 11–14% lower body weight than faLA. The oral glucose tolerance and total body fat were unchanged, but faEPA had less mesenteric fat. faEPA and faDHA had fewer large adipocytes compared to faLA and faALA. EPA reduced macrophages in the adipose tissue of *fa*/*fa* rats compared to ALA and DHA, while faLA had the greatest macrophage infiltration. DHA decreased (~10-fold) T-cell infiltration compared to faBASE and faEPA, whereas faALA and faLA had an ~40% increase. The n3-PUFA diets attenuated tumour necrosis factor-α in adipose tissue compared to faBASE, while it was increased by LA in both genotypes. In conclusion, EPA and DHA target different aspects of inflammation in adipose tissue.

## 1. Introduction

Changes in dietary and lifestyle habits over the past few decades have been linked to increases in obesity and obesity-related consequences. While obesity is physically characterized by greater adipose mass, it is adipocyte dysfunction due to the enlargement of adipocytes and enhanced triglyceride storage, which leads to an imbalance in the production of pro- and anti-inflammatory adipokines and the subsequent development of low-grade chronic inflammation, characteristic of obesity [1,2,3]. The presence of chronic inflammation contributes to the pathogenesis of several associated chronic diseases including metabolic syndrome, cardiovascular disease, type 2 diabetes mellitus, and various cancers [1,2,4]. Both dietary lipid types [saturated (SFA), monounsaturated (MUFA), and polyunsaturated fatty acids (PUFA)] and PUFA sub-types (n6 and n3) are implicated in the management of obesity and chronic inflammation associated with metabolic disorders [5,6,7,8]. Dietary recommendations for n6-PUFAs and n3-PUFAs for the general population are based on the dietary essential fatty acids [linoleic acid (LA, C18:2n6) and α-linoleic acid (ALA, C18:3n3)], whereas guidelines for disease prevention and management promote marine sources of n3-PUFAs, such as fatty fish or fish oil, containing a combination of eicosapentaenoic (EPA, C205:n3) and docosahexaenoic acids (DHA, C22:6n3) [9,10]. The reported rates of ALA elongation and desaturation to EPA and DHA based on blood measurements are low [11,12]; however, mammals fed ALA as the only dietary n3-PUFA have substantial tissue concentrations of EPA and DHA [13], and humans consuming plant-based diets without pre-formed EPA and/or DHA have no detrimental effects to their health and cognitive development [14]. Concerns regarding the sustainability of fisheries, including fish stocks and fish farming, as well as minimizing environmental impacts and exposures to ocean pollutants [15,16,17], have led to renewed questions about the effectiveness of ALA, a plant-sourced n3-PUFA present in seed oils and plants, for the prevention and management of chronic diseases.

It has been recognized that obesity is associated with lower plasma levels of n3-PUFAs [18]. Furthermore, an increased intake of n3-PUFAs has been shown to reduce obesity, especially if accompanied by a lifestyle or pharmacological intervention [19,20]. Conversely, high intakes of n6-PUFAs have been linked to greater adiposity [21]. However, these general actions of PUFAs show much greater complexity when individual fatty acids are compared, and the specific effects of plant and marine n3-PUFAs are less clear for how they impact adipose tissue and the interplay between adipocytes and the immune cells present in adipose tissue.

Adipocyte dysfunction, characterized by enlarged adipocytes and a greater production of pro-inflammatory molecules, is a key feature of the inflammatory milieu associated with obesity and metabolic disease [2,22,23,24]. Both adipocytes and infiltrating immune cells contribute to the pro-inflammatory environment within adipose tissue. Enlarged adipocytes produce more leptin, a pro-inflammatory hormone involved in appetite control, and less adiponectin, an adipokine involved in metabolic regulation and cardioprotection. Increased levels of tumour necrosis factor- α (TNF-α), a pro-inflammatory molecule, stimulate adipocytes to secrete monocyte chemoattractant protein-1 (MCP-1), leading to macrophage recruitment in adipose tissue [25,26]. Enhanced T cell activation and tissue infiltration also contribute to inflammation via the production of pro-inflammatory molecules [27,28]. When the triglyceride storage capacity of adipose tissue is exceeded, ectopic lipid accumulation in other organs contributes to further inflammation and insulin resistance [22,29].

There is a body of evidence supporting the anti-inflammatory effects of n3-PUFAs in the context of obesity and the components of adipose dysfunction described above [19,30]; however, the majority of these studies have employed fish oil as a combined source of EPA and DHA, with fewer studies investigating EPA and DHA separately or addressing the effects of the plant-based n3-PUFA ALA. Furthermore, labelling n6-PUFAs as pro-inflammatory in comparison to n3-PUFAs is an over-simplification [31,32]. Thus, there is an incomplete picture of how plant n3-PUFAs versus marine n3-PUFAs, EPA vs. DHA, and each of the n3-PUFAs versus n6-PUFAs impact inflammation in obesity. Studies delineating the effects of specific PUFA types require robust experimental designs where the proportions of SFA, MUFA, and PUFAs in the diet are similar, and only the content of the n3- and n6-PUFAs is changed. The choice of animal model requires consideration, as the response to n3-PUFA supplementation in the context of obesity induced with high-fat diets (45–60% energy) rich in saturated fat may be different from diets with low to moderate fat, given that humans develop obesity from caloric excess with a range of macronutrient distributions, including low fat and high carbohydrate dietary patterns.

Therefore, the overall objective of this study was to directly compare the n3-PUFAs, plant-based ALA in flaxseed oil, and marine-based EPA or DHA in high-purity oils, versus plant-based n6-PUFAs for their effects on adipose tissue function and glucose tolerance by assessing the adiposity, circulating adipokines, adipocyte size, and markers of inflammation and immune cell infiltration in adipose tissue as well as oral glucose tolerance in *fa*/*fa* Zucker rats. The *fa*/*fa* Zucker rat model was employed because these animals develop obesity and insulin resistance on low-fat diets (<10% *w*/*w* fat or <25% energy from fat) and are responsive to various dietary interventions [33,34]. The obese *fa*/*fa* Zucker rats also develop hepatic steatosis, and the liver-related parameters for this study have been previously reported [35].

## 2. Materials and Methods

### 2.1. Experimental Design

Five-week-old male *fa*/*fa* Zucker and lean (*+*/?) Zucker rats (Charles River Laboratories, St-Constant, PQ) underwent a minimum 1-week acclimation period. The rats were fed a diet based on the AIN-93G diet containing soybean oil [36] during the acclimation period. The *fa*/*fa* Zucker rats were randomly assigned (*n* = 10 rats/group) to the baseline group (faBASE; tissue collections at the end of the acclimation period) or to an 8-week intervention. During the intervention, *fa*/*fa* Zucker rats were fed diets containing n3-PUFAs from ALA (faALA), EPA (faEPA), or DHA (faDHA), or n6-PUFAs from LA (faLA). The lean Zucker rats were fed a diet containing n6-PUFAs from LA (lnLA) for 8 weeks. The lean Zucker rats served as a healthy reference group to determine whether interventions in the *fa*/*fa* groups were in the direction of improvement towards healthy values. Table 1 shows the diet formulations and the fatty acid compositions of the diets (as analyzed using gas chromatography). The diets contained 10% (*w*/*w*) total fat and were formulated with oil mixtures to keep the SFA, MUFA, and PUFA contents similar for all diets and to provide a similar proportion of SFAs, MUFAs, and PUFAs as in the AIN-93G semi-purified diet for rodents [36]. A dose of 3% (*w*/*w*) EPA or 3% (*w*/*w*) DHA was selected to avoid potential complications associated with higher doses [37] and was achieved using purified EPA or DHA oil (>95% purity) in free fatty acid form. ALA was provided at a similar dose (5% *w*/*w*), primarily from flaxseed oil (in triglyceride form), and combined with canola oil to achieve a similar proportion of SFAs, MUFAs, and PUFAs as the other diets. LA was provided with high linoleic safflower oil. Rats were singly caged and provided free access to the diets. Feed intake, corrected for spillage, and weekly body weights were recorded for all groups and are reported elsewhere [35]. All animal care procedures were approved by the University of Manitoba Animal Care Committee (Protocol 12-050) and conducted according to guidelines of the Canadian Council on Animal Care. The personnel preparing the diets, feeding the rats, and handling the rats for in vivo measurements were not blinded to the group allocations. The personnel analyzing the blood and tissues samples and the adipose and pancreas sections were blinded to the group allocations until after the statistical analyses were completed.

### 2.2. Body Composition and Fat Pad Weights

The total body fat was measured using whole body MRI with an EchoMRI-700™ whole body Quantitative Magnetic Resonance (QMR) instrument (Echo Medical Systems, Houston, TX, USA) and expressed relative to the body weight. At the end of the acclimation period (faBASE) or 8-week dietary intervention (experimental groups), the rats were fasted overnight and euthanized by carbon dioxide asphyxiation followed by cervical dislocation. The trunk blood was collected. The various adipose tissue pads were weighed, and portions of epididymal fat were immediately frozen in liquid nitrogen and stored at −80 °C or embedded in cryogenic gel and frozen in a dry ice–ethanol bath and stored at −80 °C. The total visceral fat was calculated as the sum of the epididymal, perirenal, and mesenteric fats. The subcutaneous fat was determined by subtracting the visceral fat from the total fat.

### 2.3. Oral Glucose Tolerance Testing

Oral glucose tolerance testing (OGTT) was completed on the baseline group during the baseline week and on the experimental groups during week 8. The rats were fasted for 5 h, and an initial blood sample was collected from the saphenous vein (t = 0). An oral glucose load (1 g glucose/kg body weight, provided as a 70% dextrose solution) was administered orally using a plastic syringe, followed by blood collection at 15, 30, 60, and 120 min post-glucose consumption. The areas under the curve (AUCs) for glucose and for insulin were calculated using the trapezoidal method. 

### 2.4. Serum Biochemistry

The serum was analyzed for glucose using a spectrophotometric assay (Genzyme Diagnostics P.E.I. Inc., Charlottetown, PE, Canada), and for insulin, leptin, adiponectin, MCP-1, and TNF-α, using singleplex immunoassays via electrochemiluminescence detection (Meso Scale Discovery, Rockville, MD, USA).

### 2.5. Pancreatic Islet Area

The pancreas was fixed in phosphate-buffered formalin (Fisher Scientific, Mississauga, ON, Canada) prior to being embedded in paraffin and sectioned. The pancreas sections were incubated with CYTO Q Background Buster (Innovex Biosciences, Richmond, CA, USA), immunostained with a monoclonal mouse anti-insulin antibody (clone E2/E3; Innovex Biosciences) followed by a peroxidase–streptavidin antibody, and then visualized with a 3,3′-diaminobenzidine tetrahydrochloride (DAB) chromogenic solution [all from the Stat Q kit, Innovex Biosciences] and counterstained with 3% hematoxylin. The images were captured with a Zeiss Axiocam digital camera using Axio Vision 4.6 (Zeiss, Thornwood, NY, USA), and the islet area was quantified using ImageJ software (Version 1.50; National Institute of Health, Bethesda, MD, USA) [38].

### 2.6. Adipocyte Size

The digital images of unstained frozen sections (10 µm) of epididymal adipose tissue were captured using Axio Vision 4.6 (Zeiss, Thornwood, NY, USA). For each group, the area of 125 adipocytes, 25 per section, was measured with Image J [38]. A circle imposed on an image of an 0.01 mm micrometer was used to calculate a conversion factor in Image J to determine the area in units of μm^2^. The adipocyte size distribution was assessed by plotting the number of adipocytes within specified size ranges for each diet group.

### 2.7. Western Immunoblotting

Western blotting was performed as described previously [39] using 10 μg of epididymal adipose tissue lysate protein. The samples were transferred electrophoretically to polyvinylidene difluoride membranes after separation in an SDS-polyacrylamide gel. Immunoblotting was subsequently performed with primary antibodies [CD3 (ab5690), F4/80 (ab74383), and MCP-1 (ab251240) from Abcam, Waltham, MA, USA; TNF-α (sc1351) from Santa Cruz Biotechnology, Dallas, TX, USA] diluted in TBST (Tris-buffered saline with Tween-20; 50 mM Tris–HCl pH 7.4, 150 mM NaCl and 0.05% Tween 20) containing 3% bovine serum albumin (BSA). Secondary horseradish peroxidase-conjugated antibodies were used at a dilution of 1:10,000 in TBST containing 1% BSA. The relative band intensities were quantified using scanning densitometry with a model GS-800 Imaging Densitometer (Bio-Rad Laboratories, Hercules, CA, USA) and normalized based on protein loading as determined by Ponceau staining of the blot.

### 2.8. Statistical Analyses

The data were analyzed using a one-way ANOVA or repeated measures ANOVA (SAS Version 9.2, SAS institute, Cary, NC, USA) for the endpoint and time course data, respectively, followed by Duncan’s multiple range test for post hoc testing. The data that were not normal or homogeneous after log transformation were analyzed using non–parametric testing with the Kruskal–Wallis test and then the least significant differences with Tukey’s correction for multiple comparisons. The Chi-squared test was used to analyze the adipocyte size distribution. Outliers (≥2.5 standard deviations from the mean) were removed from the dataset before analysis. The data were reported as the mean ± the standard error of the mean (SEM), and differences were considered significant at *p* < 0.05.

## 3. Results

### 3.1. Body Weight and Fat Accumulation

By 5 weeks of age and before being placed into their experimental groups, the *fa*/*fa* rats were already significantly larger (approximately 1.5-fold) than the lean control rats (Figure 1). Over the 8 weeks of the study, this difference remained constant. Within the *fa*/*fa* groups, however, differences in weight gain were observed (Figure 1). Specifically, the faLA group gained the greatest amount of weight, whereas the faDHA group gained the least. Although the body weight of the faALA group was not different from the faLA group throughout the study, the faALA group weighed more than the faDHA group from weeks 2 to 8. In contrast, the faEPA rats were significantly different from the faLA group from weeks 2 to 8 and similar to the faDHA group throughout the study. It can therefore be concluded that the consumption of diets containing marine-derived fatty acids resulted in a lower body weight compared to *fa*/*fa* rats fed the diets containing plant-derived fatty acids.

The differences in the body weight between lean and obese rats can be largely attributed to differences in the amount of body fat (Figure 2A). Even at baseline, the 5-week-old *fa*/*fa* rats had almost four times the percentage of total body fat compared to the lean rats at the end of the study period. Within the dietary groups, there were significant increases of 15–17% in the total body fat from baseline, except for the faDHA group, which had a non-significant 12% gain in total body fat from baseline (Figure 2A). Examining the individual fat pads indicated that there was significantly more visceral fat in the diet groups that showed the largest increases in body fat (Figure 2B). Interestingly, the reduced amount of visceral fat in the faDHA group appears to be due to smaller mesenteric fat pads rather than changes in the epididymal or perirenal fat pads (Figure 2C–E). In fact, the amount of mesenteric fat in the faDHA group was 16% less than in the baseline group. No differences in the subcutaneous fat were observed among the diet groups (Figure 2F), although it was significantly elevated at baseline in *fa*/*fa* rats relative to the lean animals.

### 3.2. Adipocyte Size and Function

Visual examination of the adipose tissue images suggested that the adipocyte size may be larger in the faLA group than in the lnLA group (Figure 3A). However, a quantification of the adipocyte size indicated there were no significant differences in the average adipocyte size among the experimental groups, despite the faLA and faALA groups having a 2.5-fold greater average adipocyte size relative to the lnLA rats (Figure 3B). Rather, the data revealed that there was greater variation in the cell size in the faLA and faALA groups relative to the other groups, and this variation in cell size was detected when the distribution of adipocyte size was examined (Figure 3C). Chi-squared testing indicated that differences among the dietary groups were present for adipocytes with areas that were <500 µm^2^, in the 2500–3000 µm^2^ range, and >3000 µm^2^ (Figure 3C). As expected, the faBASE group had 25% of its adipocytes in the smallest size range (<500 µm^2^) compared to 17% for faEPA and faDHA, 10% for lnLA and faALA, and 6% for faLA after 8 weeks of diet exposure. When the same comparison was made for the largest adipocytes, the faLA group had 20% of its adipocytes in the largest size range (>3000 µm^2^) compared to 12% for faALA and 3–6% for the remaining groups (Figure 3C). It may be concluded that some adipocytes became exceptionally enlarged in obese rats fed the LA and ALA diets, but the diet did not affect all of the adipocytes similarly.

### 3.3. Glucose Handling

Obesity is usually accompanied by insulin resistance, which was assessed by performing an oral glucose tolerance test. Fasting glucose levels were not significantly different between the groups at the zero time point before the glucose load was administered (Figure 4A). At 15 min, both the lnLA and faBASE groups had peaked, although there was a significant difference between the values (lnLA = 11.0 mmol/L, faBASE = 12.8 mmol/L). All other groups peaked between 30 and 60 min, reaching approximately 14–15 mmol/L. As would be expected, since lnLA and faBASE peaked early, they also reached the time-zero values within the 120 min test period, whereas the other groups did not come down to basal levels within that time period. These results were clearly reflected in the AUC calculation for glucose (Figure 4B), which only showed statistical differences between lnLA and faBASE versus the other diet groups, but no differences among the *fa/fa* diet groups.

With respect to serum insulin concentrations, three distinct groupings were evident prior to the consumption of the glucose load (Figure 4C). The lnLA and faBASE groups had low levels of fasting insulin (<100 µU/mL), while faLA and faDHA were moderate (400–500 µU/mL) and faALA and faEPA were high (750–800 µU/mL). No change was evident in the response to the glucose load for any group. On the other hand, the faALA group had a higher AUC for insulin compared to the faLA and faDHA groups, while the faEPA group was intermediate (Figure 4D). The AUC for insulin of the faBASE group was ~15% of the *fa*/*fa* experimental groups and similar to the lnLA group. The hyperinsulinemia and elevated AUC for insulin indicate the *fa*/*fa* rats are in an insulin resistant state, which is strongly supported by the inability to handle the glucose load as shown by Figure 4A,B.

### 3.4. Pancreas

To further examine the relationship between the diets and glucose handling, the pancreatic islets were visualized by staining sections (brown) for insulin (Figure 5A). A visual inspection revealed that the islet size varied among the groups, with the lnLA and faBASE groups having smaller islets, and the faLA, faALA, faEPA, and faDHA groups having the largest islets. This distinction was confirmed through a quantitative analysis of the islet area versus the pancreatic area (Figure 5B). These data established that the smallest islets were in the lnLA group, while the islets of the faBASE group were 5× larger. The islet size of the remaining groups was considerably greater (2× vs. faEPA–3.5× vs. faLA, faALA, and faDHA) than the control faBASE group. Interestingly, these values do not explain the variation in serum insulin levels associated with each of the diet groups (Figure 4C). This comment applies equally to the relative pancreas weight, which is less in all of the obese groups after the dietary intervention (Figure 5C). 

### 3.5. Adipose Function

To determine the functional state of the adipose tissue, select circulating adipokines were measured. Serum leptin is typically present in direct proportion to the amount of adipose tissue, and thus, it is increased in obesity. Serum leptin was elevated in all *fa*/*fa* groups, except in faEPA, compared to lnLA (Figure 6A). The lower level of leptin was observed even though the amount of adipose tissue in the faEPA group was not different from that of the other groups (Figure 2). In contrast, neither serum resistin nor adiponectin levels differed among the groups (Figure 6B,C). Adipose tissue is the major source for serum MCP-1 [40], an adipokine related to inflammation and adipose tissue dysfunction, and it was unchanged among the groups (Figure 6D). Interestingly, TNF-α was not detected in the sera of the experimental animals.

### 3.6. Adipose Inflammation

The content of key pro-inflammatory proteins in adipose tissue was examined using Western blotting (Figure 7A). Although the MCP-1 level in adipose tissue was unchanged (Figure 7B), TNFα was detected, and its levels were found to differ among the groups (Figure 7C). The lowest levels were present in the faBASE group, whereas the highest levels were found in the LA-treated animals, both lean and obese. Those on the omega-3 fatty acid diets (faALA, faEPA, and faDHA) had intermediate levels between lnLA/faLA and faBASE. The macrophage content of the tissue was assessed using F4/80 as the marker (Figure 7D). The lowest levels were found in the lnLA, faBASE, and faEPA groups. The faLA group had the highest amount of F4/80, while faALA and faDHA were intermediate. Since it has previously been reported that obesity leads to greater T cell levels in adipose tissue [41,42], we examined protein levels of the T cell marker CD3 (Figure 7E). CD3 was high in all obese groups except faDHA, in which it was even lower than that of the lnLA group. The CD3 content of faBASE was higher than lnLA as would be expected for greater obesity. However, it was noted that in all cases, except in the faBASE group, the major CD3 band had a higher molecular mass. This finding suggests that the phosphorylation state of the protein may be affected by the diet, even in lean animals. Regardless, DHA has a significant effect on the T cell content of adipose tissue, suggesting that this fatty acid reduces adipose tissue infiltration by these cells. 

## 4. Discussion

The principal outcome of the present study is the differential effects of plant n6-PUFAs, plant n3-PUFAs, and marine n3-PUFAs on body weight and adipose function as indicated by adipocyte size and markers of immune cell infiltration and inflammation in obese *fa*/*fa* Zucker rats. The DHA-enriched diet reduced the body mass of obese *fa*/*fa* Zucker rats relative to diets containing LA or ALA by selectively interfering with mesenteric fat expansion. The diet containing EPA also reduced the body mass in relation to the LA diet, but this did not occur through changes in the fat pad size. However, both faDHA and faEPA groups had a reduced number of large adipocytes in the epididymal fat, which may explain how weight reduction was achieved, whereas faLA and faALA rats had double the number of adipocytes larger than 2500 µm^2^ compared to all other diet groups. Glucose utilization in response to a glucose load was similar in all *fa*/*fa* diet groups as indicated by OGTT, whereas faLA and faDHA groups had a lower AUC for insulin compared to the faALA rats. Likewise, fasting hyperinsulinemia was ~50% lower in the faDHA and faLA groups relative to other PUFA interventions, but this was not reflected in the pancreatic islet size, which was generally the same for all diet groups. The key finding in adipose tissue was that DHA markedly reduced T cell (CD3) infiltration, while EPA prevented an elevation in macrophage (F4/80) infiltration, indicating that DHA and EPA are capable of influencing different aspects of the inflammatory response induced in adipose tissue in response to an accumulation of fat mass. The n3-PUFA diets attenuated the elevation of TNF-α (pro-inflammatory marker) in the adipose tissue of *fa*/*fa* rats compared to baseline, while the n6-PUFA diet increased TNF-α regardless of genotype. Overall, it appears that the biological effects of consuming n3-PUFAs are related to the chain length (ALA vs. EPA and DHA), while the n6-PUFA (LA) has some distinct effects related to inflammation and circulating insulin.

In the present study, both marine n3-PUFA groups weighed 11–14% less than the plant n6-PUFA group, and the faDHA rats weighed 11% less than the faALA rats (Figure 1); however, these differences were not explained by body fat, except for less mesenteric fat in the faDHA group (Figure 2). The faDHA group had less epididymal fat compared to faLA rats; otherwise, the combination of fat depots and various organs, including liver [35], having lower but non-statistically significantly different values appears to explain the lower body weight of the marine n3-PUFA groups compared to the plant n6-PUFA group. Overall, DHA had a more potent effect than EPA on the body mass and fat mass, and this concurs with other observations in the literature, whereby DHA but not EPA reduced the fat mass, but not the body weight, of mice fed high-fat diets (45% energy from fat) [43]. However, context may be an important factor to keep in mind, since it has also been reported that EPA is more effective than DHA and ALA in reducing fat mass, without changes in the body weight, in C57Bl/6 mice fed high-fat, high-sucrose diets [44]. However, this beneficial effect of EPA on adiposity was not observed in genetically obese *ob*/*ob* mice [44]. The lack of effectiveness of an EPA treatment with *ob*/*ob* mice is surprising given the results we obtained with *fa*/*fa* rats, where both EPA and DHA lowered body mass, since both mutations interfere with the regulation of satiety by leptin, thus leading to a similar hyperphagic phenotype [45]. Interestingly, the marine n3-PUFAs attenuated the body weights in the present study, in the context of low-fat diets without changes in feed intake or feed efficiency [35], of *fa*/*fa* rats that were obese soon after birth [46]. In mice, EPA was effective for reducing body mass and adiposity during concomitant high-fat feeding, but not when EPA was provided to mice with established diet-induced obesity [47]. In contrast to these purported actions of DHA and EPA, there is little evidence to support a positive effect by ALA on body weight or adiposity based on the current study or our previous publication [41], likely due to the limited conversion to EPA and DHA [48] and reduced amounts of EPA and DHA in key tissues [13]. While these findings indicate that ALA may not impact adiposity, ALA has been shown to function in other circumstances independent of its conversion [49]. Indeed, changing the n6:n3 ratio by the addition of ALA to the diet did not result in a change in the body mass in the current study or a previous study with *fa*/*fa* rats [41]. However, our observations are in accordance with those of Pinel et al. [44], which suggest that DHA and EPA operate by different mechanisms [50], with the fat-reducing ability of DHA possibly due to the induction of lipolysis [51] and/or inhibition of adipose tissue angiogenesis by a Sirt-1-dependent mechanism [52].

Two observations stand out with respect to the fat depots and adipocyte size. First, DHA selectively affects the mesenteric fat tissue, decreasing it below baseline levels, but it was unaffected by EPA treatment (Figure 2). Second, both DHA and EPA prevented an increase in the number of extremely large adipocytes (Figure 3) that are typical of obesity [2,22]. Others have reported smaller adipocytes in subcutaneous and epididymal fat with EPA or DHA supplementation of mice with diet-induced obesity [47,52,53,54] but have not directly compared EPA and DHA in the same study. While we cannot explain the selectivity of DHA to a single fat pad in the present study, retaining small adipocytes would ensure a strong positive effect on their metabolic characteristics. Indeed, small adipocytes would function normally with respect to lipid storage, largely due to their ability to respond to metabolic hormones, as well as their ability to communicate with other metabolically active tissues via adipokine secretion [2,22]. Surprisingly, the serum leptin, adiponectin, and resistin concentrations (Figure 6) were unaffected by variations in the fat mass or adipocyte size in *fa*/*fa* rats fed n3-PUFA diets, suggesting that another mechanism may be responsible. In a previous study of diet-induced obesity in rats (55% energy from fat), the MUFA content of the diet rather than the size of the animals appeared to have the greatest effect on adiponectin levels in adipose tissue, whereas circulating leptin was unchanged [55]. This could explain the lack of change seen in the current study regarding adiponectin, since the MUFA content of the intervention diets was kept constant. In the present study, the serum leptin of the faEPA group was not different from lnLA rats (Figure 6), suggesting an improvement; however, this was not explained by fat pad weights or adipocyte sizes of the faEPA rats relative to the other *fa*/*fa* groups. Others have reported that EPA supplementation reduced circulating leptin in mice fed a Western diet but not in genetically obese *ob*/*ob* mice compared to non-supplemented obese controls [44], whereas there were no differences in the plasma leptin levels when humans received supplements containing only EPA [56]. With respect to other cytokines present in adipose tissue, it has been reported that TNF-α may contribute to hypertrophic growth by suppressing adipogenesis [57,58]. Thus, the reduction in TNF-α levels in the adipose of faEPA and faDHA groups may explain why fat pads in these groups have smaller adipocytes; however, this does not explain why an ALA treatment did not result in a similar outcome as the marine n3-PUFA. The adipocyte size of faALA was similar to faLA, unlike previous reports of smaller adipocytes when a flax oil-rich diet was provided to high-fat fed rats or older *fa*/*fa* rats [41,59].

The key finding in adipose tissue was the distinct effects of EPA for reducing macrophages and DHA for decreasing T cells, while LA increased both of these immune cell types. EPA reduced macrophage levels by ~50% in *fa*/*fa* rats compared to ALA and DHA and to a level equivalent to lean rats, while faLA had the highest macrophage infiltration based on the macrophage marker F4/80 [60]. Other studies focusing solely on EPA or DHA have reported less macrophage infiltration (based on immunostaining for galactin-3 or F4/80) in the adipose tissue of mice during obesity development with high-fat diets [52,61], whereas the present study reveals that EPA is more effective than DHA or ALA for preventing increased macrophage infiltration. These findings were not explained by MCP-1, a key factor in the recruitment of monocytes to dysfunctional adipose tissue [25,62], as the circulating and adipose tissue levels of MCP-1 were unchanged (Figure 6 and Figure 7). Perhaps other regulators of macrophage recruitment are involved, such as macrophage migration inhibitory factor (MIF), which has been investigated in the context of MIF^-/-^ mice and high-fat diets [63] but not for the effects of n3-PUFA diets. Furthermore, recent data are demonstrating the importance of cyclooxygenase-2 (COX-2) in monocytes/macrophages for the increased recruitment and proliferation of adipose tissue macrophages, and a role for prostaglandin E2 (PGE2) and its receptor (EP4) on macrophages [64]. That study employed genetic (selective deletion of myeloid cell COX-2 or EP4) or pharmacological approaches (EP4 agonist) [64] to manipulate COX-2 and PGE2 levels. Arachidonic acid, the preferred COX substrate [65], produces PGE2; however, higher amounts of EPA will reduce PGE2 and lead to more of the EPA-derived metabolites [66]. Furthermore, macrophage differentiation occurs in response to PGE2 produced by mesenchymal stromal cells [67]. Based on these findings, it is possible to speculate that in the present study, the EPA diet may be reducing macrophages in adipose tissue via a signaling pathway involving COX-2, a reduction in PGE2, and the effects of fatty acid metabolites (oxylipins) produced from EPA such as PGE3. This requires further investigation given the evidence that an n3-PUFA treatment in vitro alters the oxylipin profile of M1- and M2-like macrophages [68,69] and that adipocytes are responsive to oxylipins [70], yet the effects of oxylipins produced from EPA (or other PUFAs) by COX on the interplay between monocytes/macrophages and adipocytes are largely unknown.

In contrast to the effects on macrophages, the DHA intervention resulted in an ~10-fold decrease in T cell (CD3) infiltration compared to *fa*/*fa* rats at baseline, and to a level statistically equivalent to the lnLA control. EPA maintained CD3 at baseline levels, whereas the faALA and faLA groups had an ~40% increase in T cell infiltration compared to baseline. In older *fa*/*fa* Zucker rats provided a high ALA diet, we had previously reported a reduction in the CD3 protein in epididymal fat [41]. The growing recognition of lymphocyte (T cell) infiltration into adipose tissue in obesity and the role of adaptive immunity in obesity-induced inflammation [27,71,72] indicates the need for future studies to employ flow cytometry to further distinguish the effects of n3-PUFAs on the infiltration of specific T-cell subtypes, since others have shown increased total T cells (CD3^+^), T helper cells (CD3^+^CD4^+^), and cytotoxic T cells (CD3^+^CD8^+^) in the visceral adipose tissue of mice with diet-induced obesity [42]. Cucchi et al. [73] showed that fish oil supplementation in lean C57BL/6 mice redistributed the CD4^+^ T-cell subsets in fat and lymphoid tissues in favour of an anti-inflammatory phenotype, and that reduced T cell motility may be involved. These concepts now need to be tested with specific n3-PUFAs in the context of obesity. Another observation that stood out was the high degree of CD3 phosphorylation in all intervention groups relative to faBASE (Figure 7A). While the phosphorylation of CD3 is indicative of T cell activation [74], there is little information in the literature to provide a foundation for speculating what this modification means in the context of obesity. Perhaps further investigations will support CD3 phosphorylation as an alternative marker for activated lymphocytes (akin to CD3^+^CD4^+^CD25^lo^ cells) in adipose tissue when flow cytometry is not employed [42]. Also, further exploration of this observation will be necessary to identify the relationship between the fatty acid composition of a diet, the activation of CD3, and the roles of specific T-cell subsets, especially as they relate to the immunosuppressive actions of n3-PUFAs in the context of obesity. 

The ability of DHA and potentially EPA to suppress adipocyte dysfunction may form the basis for the beneficial actions of these n3-PUFAs to prevent and/or reverse fatty liver disease, as we previously reported for these rats [35]. This statement relates directly to a recent publication that provides strong evidence for adipose tissue dysfunction as a causal factor in the development of NAFLD [3] and the subsequent development of insulin resistance [3,75,76]. Not only do these fatty acids decrease the size of adipocytes, they help to retain their normal functionality as indicated by reduced immune cell infiltration. As a result, lipid metabolism is improved, inflammation declines, and the development of fatty liver disease and its concomitant insulin resistance are reversed in the context of dietary EPA or DHA intervention in obese rats [35]. On the other hand, the improvements in adipose function were not associated with changes in the oral glucose tolerance or pancreatic islet area of *fa*/*fa* rats despite a 50% reduction in fasting hyperinsulinemia and a smaller AUC for insulin with DHA or LA consumption. In fact, the changes in insulin-related parameters, including less insulin resistance as indicated by the homeostasis model assessment of insulin resistance [35], were the only beneficial responses noted with the plant n6-PUFA diet. This was somewhat surprising given that the faLA group had the highest levels of pro-inflammatory marker TNF-α in adipose tissue, and pro-inflammatory markers such as TNF-α are associated with insulin resistance and systematic inflammation [77]. Interestingly, both lean and *fa*/*fa* rats fed the LA diet had elevated TNF-α in adipose tissue, suggesting that the effects of the n6-PUFA diet were dominant over genotype. 

A strength of this research is the comparison of the three major n3-PUFAs separately (ALA vs. EPA vs. DHA) at the same dose and with isocaloric diet formulations that kept the proportions of SFAs, MUFAs, and PUFAs constant. The study design enabled the direct comparison of EPA versus DHA by employing high-purity EPA oil and DHA oil, whereas many other studies examine the effects of fish oil containing varying combinations of EPA and DHA. Furthermore, incorporating fish oil in a diet does not necessarily maintain constant proportions of SFAs, MUFAs, and PUFAs compared to the control diet. Although it was a strength to compare the three n3-PUFAs at the same dose in an eight-week basic sciences study in a preclinical model, a limitation of the present study is that these high amounts of EPA and DHA would not be realistic or safe for humans [78,79]. Another strength of this study is that the n3-PUFA diets were compared to an n6-PUFA diet in two controls: the obese *fa*/*fa* genotype and the healthy lean genotype. The examination of two different immune cell types was a strength, and the results demonstrated the distinct effects of EPA versus DHA via Western blotting of adipose tissue. A limitation was the lack of flow cytometry to examine the sub-types of macrophages and T cells; however, this procedure has to be planned and coordinated at the end of the feeding trial, as fresh adipose tissue is required to isolate the stromal vascular fraction and to label immune cells with the appropriate combinations of fluorescent antibodies for a flow cytometric analysis. The differential effects of EPA and DHA on immune cells in adipose tissue of the present study, and our previous report of the divergent effects of EPA and DHA on hepatic steatosis [35] indicate that future studies investigating the effects of EPA and DHA on immune cell subtypes need to extend beyond the adipose tissue and obesity to the liver and hepatic steatosis [80] as well as other organs relevant to chronic disease conditions.

## 5. Conclusions

The plant and marine n3-PUFAs improved adipose function to different degrees and using different mechanisms. The marine n3-PUFAs reduced the body weight and size of adipocytes of obese *fa*/*fa* rats, without changing the total body fat, compared to n6-PUFA. DHA selectively reduced mesenteric fat depots. Plant and marine n3-PUFAs had fewer macrophages in adipose tissue compared to n6-PUFAs, with EPA lowering macrophages to a greater extent than ALA and DHA interventions. Only DHA decreased T cells in adipose tissue and by ~10-fold compared to baseline. Both ALA and DHA lowered levels of the pro-inflammatory molecule TNF-α in adipose tissue compared to n6-PUFA. Further research is needed to build on the novel finding of the current study with respect to the distinct effects of marine n3-PUFAs on the macrophages (innate immune cells) and DHA on the T cells (adaptive immune cells) present in adipose tissue in the context of obesity.

## Figures and Tables

**Figure 1 nutrients-16-01311-f001:**
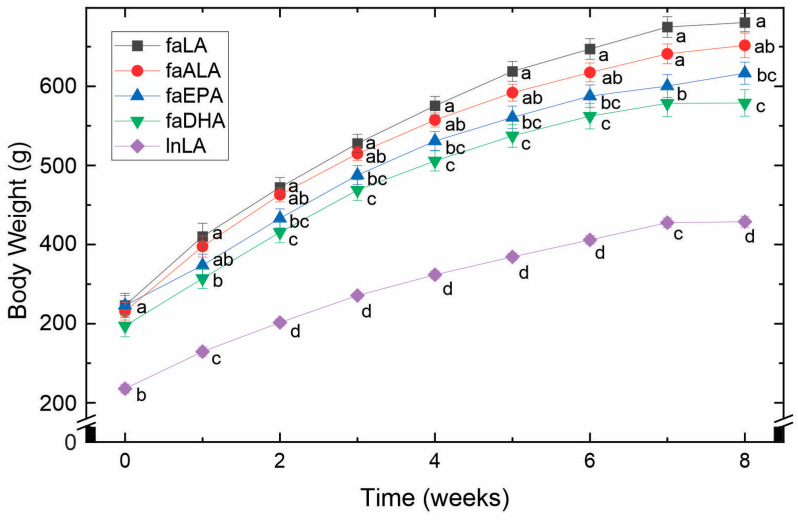
Effects of diets containing different polyunsaturated fatty acids on body weight. Weekly body weights obtained over the course of the 8-week feeding period for each diet group are shown as the mean ± SEM (*n* = 8–10). Data were analyzed using a repeated measures ANOVA, and letters indicate significant differences (*p* < 0.05) at each week based on post hoc testing with Duncan’s multiple range test. Abbreviations: faALA, *fa*/*fa* rats fed the α-linolenic diet; faDHA, *fa*/*fa* rats fed the docosahexaenoic diet; faEPA, *fa*/*fa* rats fed the eicosapentaenoic diet; faLA, *fa*/*fa* rats fed the linoleic diet; lnLA, lean Zucker rats fed the linoleic diet.

**Figure 2 nutrients-16-01311-f002:**
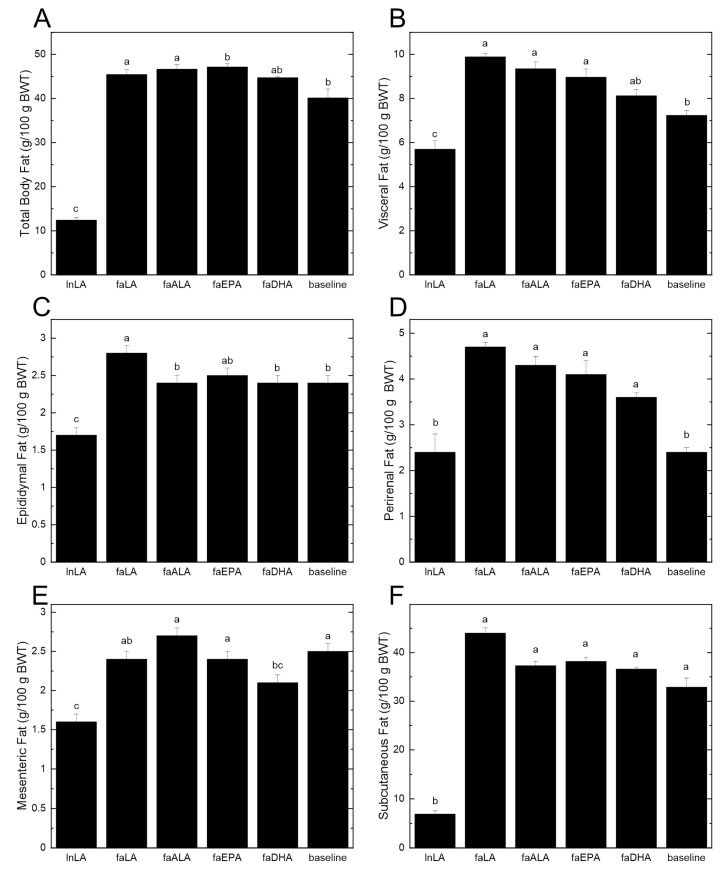
Effects of different polyunsaturated fatty acids on total body fat and individual fat pads. (**A**) Total body fat was determined using an EchoMRI-700™ whole body QMR instrument (Echo Medical Systems, Houston TX, USA). Individual fat pads were dissected and weighed at the end of the 8-week feeding period. Visceral fat (**B**) was calculated by summing the values for epididymal fat (**C**), perirenal fat (**D**), and mesenteric fat (**E**). Subcutaneous fat (**F**) was determined by subtracting visceral fat from the total body fat. All data are presented as the mean ± SEM (*n* = 8–10) relative to the total body weight (BWT). Data were analyzed using a one-way ANOVA, and different letters indicate statistically significant differences (*p* < 0.05) based on post hoc testing with Duncan’s multiple range test. Total body fat, total visceral fat, and subcutaneous fat were analyzed using the Kruskal–Wallis test, followed by least significant difference post hoc testing with Tukey’s correction for multiple comparisons.

**Figure 3 nutrients-16-01311-f003:**
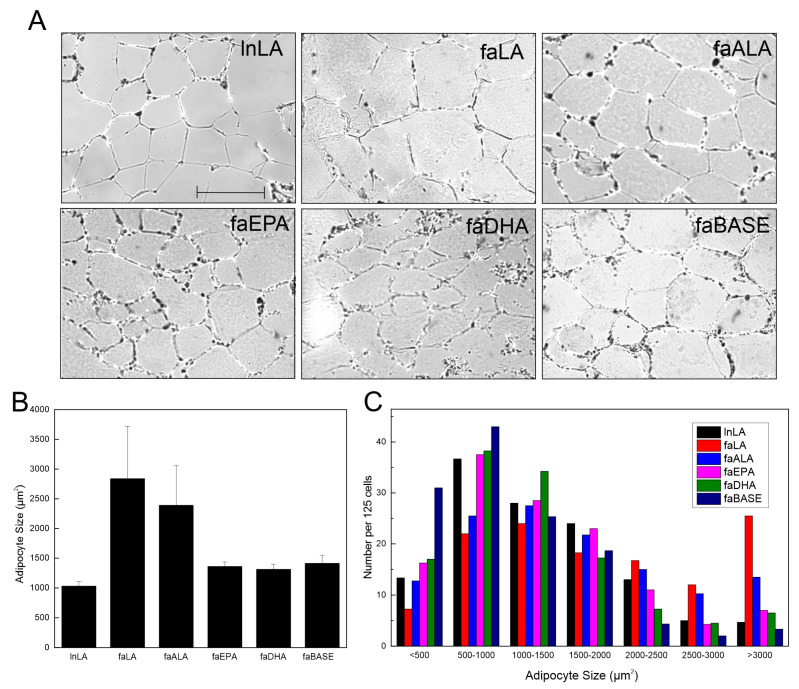
Effects of diets containing different polyunsaturated fatty acids on adipocyte size. (**A**) Representative images of adipose tissue sections for each diet group. Scale bar = 0.05 mm. (**B**) Average adipocyte size was calculated from 125 randomly selected adipocytes/animal. The data are presented as the mean ± SEM (*n* = 3–4). No statistical differences were detected between treatment conditions using a one-way ANOVA. (**C**) Distribution of adipocyte size for each treatment group within the indicated size ranges. The data are presented as the number of cells per range for 125 cells from each group. Chi-squared testing determined there were significant differences among groups within each size range.

**Figure 4 nutrients-16-01311-f004:**
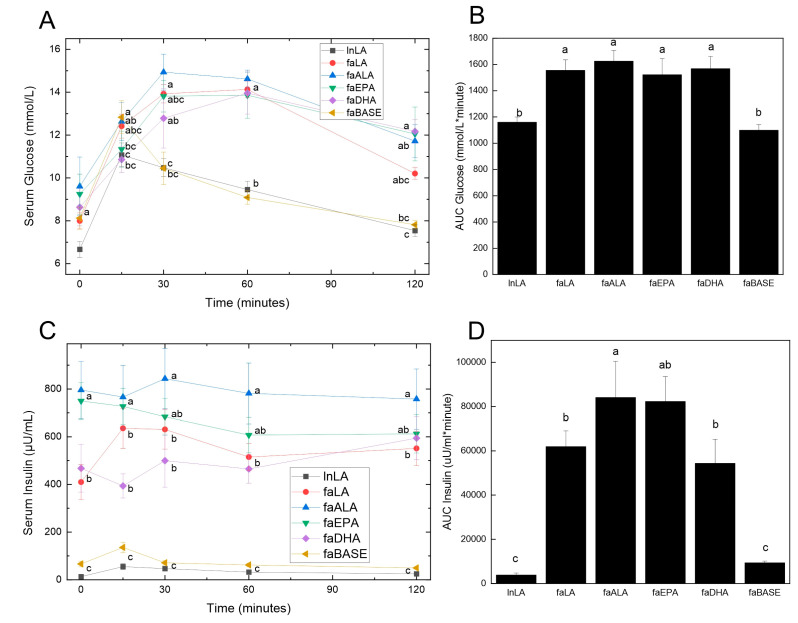
Effects of diets containing different polyunsaturated fatty acids on oral glucose tolerance. Oral glucose tolerance testing was performed during week 8 of the feeding period. Serum glucose (**A**) was measured using blood samples obtained from the saphenous vein at the indicated times, and the values were used to calculate the area under the curve (AUC) (**B**). Insulin was measured in the same blood samples obtained during the oral glucose test (**C**) and likewise used to calculate the AUC (**D**). Data are presented as the mean ± SEM (*n* = 6–10). Data were analyzed using a repeated measures ANOVA (**A**,**C**) or one-way ANOVA (**B**,**D**), and different letters indicate statistical significance (*p* < 0.05) from other values at the same time point or among groups.

**Figure 5 nutrients-16-01311-f005:**
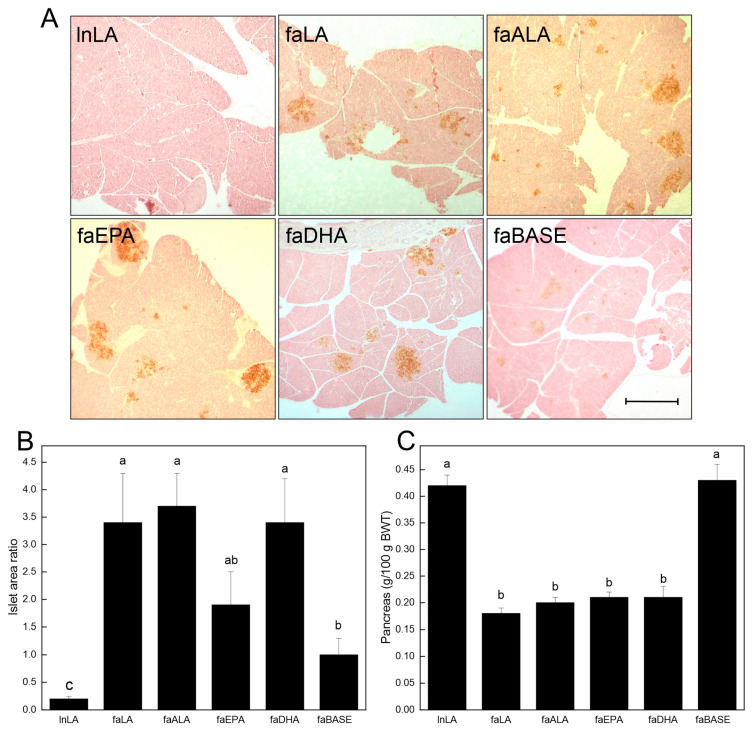
Effects of diets containing different polyunsaturated fatty acids on the pancreas. (**A**) Representative images of pancreas tissue sections for each diet group after immunostaining for insulin to visualize the islets. Scale bar = 0.05 mm. (**B**) The % islet area was calculated as the islet area of a section/total pancreas area of a section × 100, with the areas quantified using ImageJ. The data are presented as the mean ± SEM (*n* = 6–8). (**C**) The pancreas weight relative to body weight (BWT) is presented as the mean ± SEM (*n* = 9–10). Data were analyzed using a one-way ANOVA (**B**,**C**), and different letters indicate statistical significance (*p* < 0.05) among the means based on post hoc testing with Duncan’s multiple range test.

**Figure 6 nutrients-16-01311-f006:**
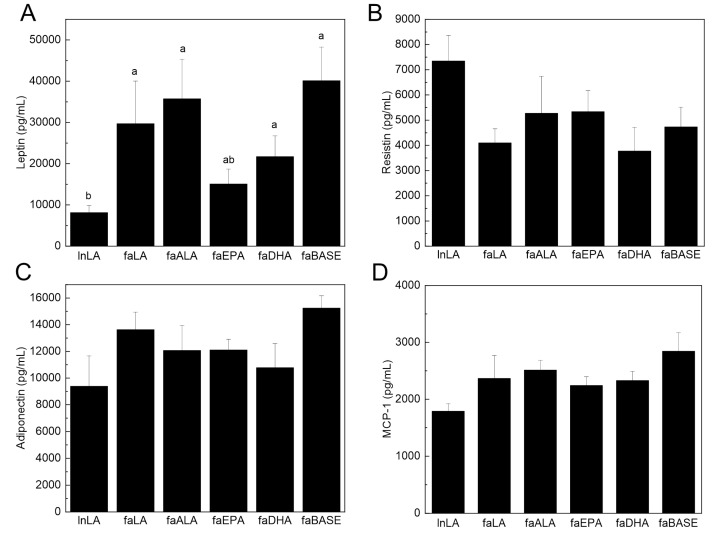
Effects of diets containing different polyunsaturated fatty acids on serum biomarkers of adipose tissue function and inflammation. Serum leptin (**A**), resistin (**B**), adiponectin (**C**), and monocyte chemoattractant protein-1 (MCP-1) (**D**) were measured using an immunoassay. The data are presented as the mean ± SEM (*n* = 5–10). The data were analyzed using a one-way ANOVA, and different letters indicate significant differences (*p* < 0.05) among the means based on post hoc testing with Duncan’s multiple range test. An absence of letters indicates no significant differences.

**Figure 7 nutrients-16-01311-f007:**
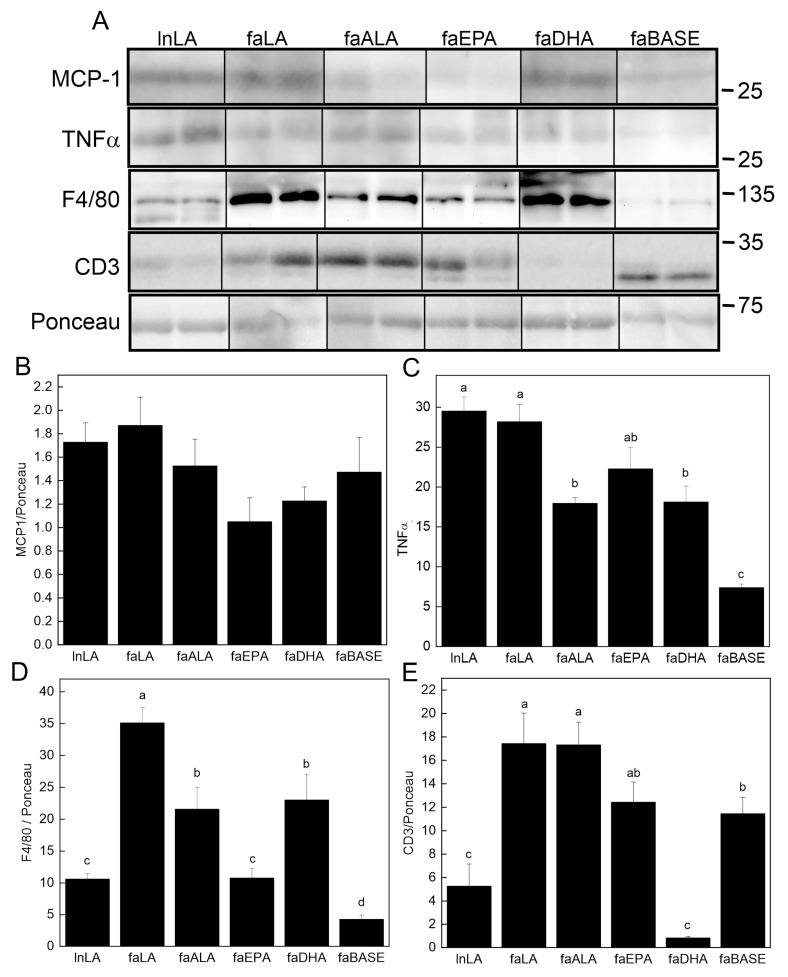
Effects of diets containing different polyunsaturated fatty acids on markers of adipose tissue inflammation. (**A**) Representative Western blots showing levels of monocyte chemoattractant protein-1 (MCP-1), tumour necrosis factor-α (TNF-α), and F4/80 relative to total protein visualized with Ponceau stain. The positions of molecular weight markers (in kDa) are indicated. Densitometry was used to quantify the bands of interest (all present on the same blot; positions were rearranged to fit the sample order used in the figures) which were normalized to the total protein levels (band visualized by Ponceau stain). The graphs show the relative band intensity as the mean ± SEM (*n* = 6) for MCP-1 (**B**), TNF-α (**C**), F4/80 (**D**), and CD3 (**E**). Data were analyzed using a one-way ANOVA, and different letters indicate significant differences (*p* < 0.05) among the means based on post hoc testing with Duncan’s multiple range test. An absence of letters indicates no significant differences.

**Table 1 nutrients-16-01311-t001:** Diet formulations.

	LA Diet	ALA Diet	EPA Diet	DHA Diet
Diet Ingredients (g/kg) ^1^				
Cornstarch	348	348	348	348
Maltodextrin	132	132	132	132
Sucrose	100	100	100	100
Egg white	213	213	213	213
Cellulose	50	50	50	50
AIN-93G-MX mineral mix	35	35	35	35
AIN-93-VX vitamin mix	10	10	10	10
Choline	3	3	3	3
Biotin mix ^2^	10	10	10	10
Soybean oil	0	0	67	67
High linoleic safflower oil ^3^	100	0	0	0
Flaxseed oil ^4^	0	87	0	0
Canola oil ^5^	0	10	0	0
Coconut oil ^6^	0	3	0	0
EPA oil ^7^	0	0	33	0
DHA oil ^7^	0	0	0	33
Fatty Acid Composition (g/100 g lipid) ^8^				
SFAs	10	11	10	10
MUFAs	17	19	15	15
PUFAs	72.3	70	75	75
LA (C18:2n6)	72	18	36	36
ALA (C18:3n3)	0.3	52	6	6
EPA (C22:5n3)	0	0	32	0
DHA (C22:6n3)	0	0	0	33
Other PUFAs	0	0	1	0
n6-PUFA:n3-PUFA	240:1	1:3	1:1.1	1:1.1

^1^ Ingredients from Dyets, Inc. (Bethlehem, PA, USA) unless otherwise indicated; diets were isocaloric and provided 3.9 kcal/gram. ^2^ 200 mg biotin/kg cornstarch was added, because egg white contains avidin, which binds to biotin. ^3^ Alnor Oil Company (Valley Stream, NY, USA). ^4^ Omega Nutrition (Vancouver, BC, Canada). ^5^ Smuckers Food Services (Markham, ON, Canada). ^6^ Nutiva (Richmond, CA, USA). ^7^ Larodan Fine Chemicals (Malmö, Sweden); >95% purity. ^8^ Determined using gas chromatography. Abbreviations: AIN, American Institute of Nutrition; ALA, α-linolenic acid; DHA, docosahexaenoic acid; EPA, eicosahexaenoic acid; LA, linoleic acid; MUFAs, monounsaturated fatty acids; PUFAs, polyunsaturated fatty acids; SFAs, saturated fatty acids.

## Data Availability

The data supporting the conclusions of this article will be made available by the authors upon reasonable request.

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
