# Peer review of "Differential Modulation by Eicosapentaenoic Acid (EPA) and Docosahexaenoic Acid (DHA) of Mesenteric Fat and Macrophages and T Cells in Adipose Tissue of Obese fa/fa Zucker Rats"

_nutrients, 2024, doi:10.3390/nu16091311_

Round 1
Reviewer 1 Report
Comments and Suggestions for Authors
In the manuscript “Differential modulation by eicosapentaenoic acid (EPA) and docosahexaenoic acid (DHA) of mesenteric fat and macrophages and T cells in adipose tissue of obese fa/fa Zucker rats”, Lena Hong, et al. Based on the current study of chronic inflammation caused by obesity and the frequent occurrence of related diseases, this paper uses fa/fa Zucker rats, a classic genotype obese rat model, to study the effects of n3-PUFA and n6-PUFA on adipose tissue function and inflammation in rats.
In general, this paper explores the effects of EPA, DHA, ALA and LA on adipocyte function and obesity-related inflammation, expands the functional research on these n3-PUFA. And the effects of n6-PUFA are not simply considered to promote inflammation, providing new ideas for alleviating obesity complications. Generally speaking, the quality is acceptable, but some of the figures still have problems and need to be carefully checked and corrected. In addition, there are many factors affecting adipocyte function and inflammation in the microenvironment of the body, and the mechanism by which n3-PUFA affects adipocytes is unknown, the quality would be improved if it could be supplemented with experiments on adipocytes.
Below are detailed comments:
1. In introduction. It is suggested to introduce the details of n3-PUFA and n6-PUFA in this section.
2. In materials and methods. Line 96. “The lean Zucker rats that served as a healthy reference group were fed a diet containing n6-PUFA from LA (lnLA) for 8 weeks”. What is the role of this group in the overall experimental design?
3. In results. Line 325-326. “Although MCP-1 in adipose tissue was unchanged, TNFα was detected and found to differ among the groups”. It is different from the situation you mentioned in the introduction (Line 55-56). So what is the key factor that lead to macrophage recruitment in fa/fa Zucker rats? Maybe it would be better to have a detailed analysis and prediction in the discussion section.
4. In results. Line 338-339. Would it be better to use flow cytometry to compare T cell changes?
Minor comments:
1. In Figure 1, the y-axis should start at 0, maybe a truncation could be done here.
2. In Figure 3 A, the specific size of the scale of each image should be labeled.
3. Figure 7 is placed inside the Discussion, which is obviously inappropriate. Please correct.
4. In figure 7 A, the figure is not very clear, please submit the original image of the relevant results.
Reviewer 2 Report
Comments and Suggestions for Authors
This study demonstrated that “Differential modulation by eicosapentaenoic acid (EPA) and docosahexaenoic acid (DHA) of mesenteric fat and macrophages and T cells in adipose tissue of obese fa/fa Zucker rats”. This manuscript is interesting, however, there are several concerns relating that should be carefully address by the authors.
Materials and Methods
How did authors decide the components of each diet, especially the amount of PUFA?
Authors should explain about it.
Reaults
The weight of total body fat of the faDHA group was decreased only a few g compared to that of faLA group. Nevertheless, the body weight between the two groups was differed almost 70 g.
What is the reason for this? Was there a difference in the weight of each organ? Authors should describe about it.
Why was serum leptin level of faEPA group low as same as that of InLa group?
Authors should describe about it.
This manuscript lacks the data on the mechanisms of different regulatory mechanisms of mesenteric fat, adipose tissue macrophages and T cells by EPA and DHA.
Authors should add the several data to reveal of these mechanisms systematically.
Round 2
Reviewer 1 Report
Comments and Suggestions for Authors
No more comments
Author Response
Thank you for your comments.
Reviewer 2 Report
Comments and Suggestions for Authors
Authors almost responded for my comments. I hope you should resolve the mechanisms of different regulatory mechanisms of mesenteric fat, adipose tissue macrophages and T cells by EPA and DHA in near future.
Author Response
Thank you for your comments.